# The Role of the Gut Microbiota in Female Reproductive and Gynecological Health: Insights into Endometrial Signaling Pathways

**DOI:** 10.3390/life15050762

**Published:** 2025-05-09

**Authors:** Patricia Escorcia Mora, Diana Valbuena, Antonio Diez-Juan

**Affiliations:** R&D Department, Igenomix (Part of Vitrolife Group), Ronda de Narcís Monturiol, nº11, B, Edificios Europark, Parque Tecnológico, 46980 Paterna, Valencia, Spain; patricia.escorcia@igenomix.com (P.E.M.); diana.valbuena@igenomix.com (D.V.)

**Keywords:** gut–endometrial axis, gut dysbiosis, implantation failure, reproductive immunology, estrogen metabolism, microbial metabolites, microbial translocation, estrobolome, gynecological disorders

## Abstract

Fertility is a dynamic, multifactorial process governed by hormonal, immune, metabolic, and environmental factors. Recent evidence highlights the gut microbiota as a key systemic regulator of reproductive health, with notable impacts on endometrial function, implantation, pregnancy maintenance, and the timing of birth. This review examines the gut–endometrial axis, focusing on how gut microbial communities influence reproductive biology through molecular signaling pathways. We discuss the modulatory roles of microbial-derived metabolites—including short-chain fatty acids, bile acids, and tryptophan catabolites—in shaping immune tolerance, estrogen metabolism, and epithelial integrity at the uterine interface. Emphasis is placed on shared mechanisms such as β-glucuronidase-mediated estrogen recycling, Toll-like receptor (TLR)-driven inflammation, Th17/Treg cell imbalance, and microbial translocation, which collectively implicate dysbiosis in the etiology of gynecological disorders including endometriosis, polycystic ovary syndrome (PCOS), recurrent implantation failure (RIF), preeclampsia (PE), and preterm birth (PTB). Although most current evidence remains correlational, emerging insights from metagenomic and metabolomic profiling, along with microbiota-depletion models and Mendelian randomization studies, underscore the biological significance of gut-reproductive crosstalk. By integrating concepts from microbiology, immunology, and reproductive molecular biology, this review offers a systems-level perspective on host–microbiota interactions in female fertility.

## 1. Introduction

Fertility is a multifactorial process influenced by genetic, hormonal, immunological, and environmental factors. Emerging research has identified gut microbiota as a critical regulator of reproductive health, exerting systemic effects on metabolism, immune function, and endocrine signaling [1]. Gut dysbiosis—defined as an imbalance in microbial communities—has been associated with a variety of reproductive disorders, including infertility, via mechanisms that involve disrupted endometrial signaling, chronic inflammation, and hormonal dysregulation [2].

The female reproductive tract also harbors its own microbiome, which interacts dynamically with gut microbial populations through immune, neural, endocrine, and metabolic axes [2]. This bidirectional communication forms the basis of the gut–endometrial axis, a physiological interface through which microbial metabolites and immune signals influence uterine function and some gynecological disorders such as endometriosis, polycystic ovary syndrome (PCOS), recurrent implantation failure (RIF), preterm birth (PTB), and unexplained infertility [3].

Mechanistically, changes in gut microbiota composition can reshape systemic estrogen levels through the estrobolome—a subset of microbial genes responsible for estrogen metabolism—thereby impacting endometrial receptivity and embryo implantation [4]. Additionally, microbial-derived metabolites such as short-chain fatty acids (SCFAs), bile acids (BAs), and tryptophan catabolites influence immune tolerance, epithelial integrity, and inflammatory tone within the endometrium [5,6,7]. These mediators interact with host receptors and signaling networks to regulate T cell differentiation, cytokine expression, and endometrial remodeling [8,9].

While therapeutic applications are still under development, understanding this gut-reproductive crosstalk lays the groundwork for interventions aimed at modulating microbial composition to restore hormonal balance and immune tolerance. As tools such as metagenomics, metabolomics, and microbiome-editing therapies evolve, the gut microbiota may become a novel target in fertility medicine [10].

This review focuses on the molecular mechanisms by which the gut microbiota influences endometrial function, with a specific focus on signaling pathways that affect implantation and reproductive outcomes. Rather than a passive bystander, the gut microbiota may be a key orchestrator of endometrial receptivity, fertility outcomes, and the development of pathological conditions.

## 2. Determinants of Gut Microbial Composition and Its Implications for Endometrial Function

The composition of the adult gut microbiota is shaped by a complex interplay of intrinsic and extrinsic factors, including age, genetic predisposition, diet, physical activity, psychological stress, and exposure to endocrine-disrupting chemicals (EDCs). These variables not only influence microbial diversity and community structure but also exert profound effects on systemic metabolism, immune function, and host reproductive health [11].

### 2.1. Microbial Trajectories Across the Lifespan: Impact on Reproductive Immunity and Estrogen Homeostasis

It has been observed that the composition of the human gut microbiome changes throughout life. Microbial colonization begins at birth and is influenced by delivery mode and early feeding. Vaginally delivered infants acquire beneficial taxa such as *Lactobacillus* and *Bifidobacterium*, which support immune maturation. In contrast, cesarean deliveries often result in delayed colonization and dysbiosis, increasing susceptibility to allergic and autoimmune conditions [12,13]. In adulthood, the gut microbiome achieves a relatively stable composition but remains sensitive to lifestyle and environmental influences, including dietary habits, medication use, and psychological stress [14,15].

With advancing age, the gut microbiota undergoes a progressive loss of diversity, termed “microb-aging”, marked by declines in beneficial Firmicutes and Bacteroidetes and an expansion of pathobionts such as *Enterobacteriaceae* [16]. This shift promotes low-grade systemic inflammation (“inflammaging”), metabolic dysregulation, impaired gut barrier function, and increased frailty [17,18].

### 2.2. Environmental Modulators of the Gut Microbiota and Their Influence on Systemic Metabolic and Immune Responses

Environmental factors such as poor nutrition, physical inactivity, and polypharmacy exacerbate age-related shifts in the gut microbiota. These changes significantly alter microbial composition and systemic outputs, with far-reaching consequences for both reproductive and overall health. To address these challenges, emerging interventions—including microbiota-targeted diets, probiotics, and fecal microbiota transplantation (FMT)—are under investigation to mitigate age-associated decline and support healthy longevity [19].

Diet is a major and rapidly acting modulator of gut microbial composition. Fiber-rich, plant-based diets enhance the growth of SCFA-producing taxa such as *Faecalibacterium prausnitzii*, *Lactobacillus*, and *Bifidobacterium*, which support mucosal barrier integrity and immune tolerance [20]. In contrast, Western-style diets, which are high in saturated fats and refined sugars, reduce microbial diversity and promote the growth of pro-inflammatory species such as *Bilophila wadsworthia* and *Alistipes*, contributing to chronic inflammation and metabolic dysregulation [18]. Additionally, polyphenols and fermented foods enrich beneficial microbes like *Akkermansia muciniphila*, which are linked to improved metabolic and reproductive outcomes [21,22].

Regular physical activity promotes microbial diversity and enhances the production of SCFAs, favoring beneficial taxa such as *Akkermansia* and *Roseburia* while suppressing inflammatory signaling [23,24]. In contrast, sedentary behavior is associated with reduced microbial richness and an expansion of Proteobacteria, impairing the gut barrier function and systemic immune responses [25,26].

Psychological stress alters gut microbial ecology through the activation of the hypothalamic–pituitary–adrenal (HPA) axis and increased cortisol secretion. This response raises intestinal permeability and elevates circulating lipopolysaccharide (LPS) levels, contributing to systemic inflammation [27,28]. Stress-related dysbiosis is characterized by a reduction in *Bifidobacterium* and an increase in Firmicutes, which are linked to mood disorders, chronic inflammation, and adverse reproductive outcomes via immune–endocrine disruption [29,30].

Exposure to EDCs—such as bisphenol A (BPA), phthalates, and certain pesticides—disturbs microbial homeostasis by selectively depleting beneficial taxa and promoting pro-inflammatory microbes. These alterations negatively impact estrogen metabolism, immune signaling, and fertility potential [2].

### 2.3. Host Genetic Control of Gut Microbiota: Relevance for Endometrial Signaling Pathways

Large-scale genome-wide association studies (GWAS) have identified multiple genetic loci that influence gut microbiota composition. However, the heritability of most bacterial taxa remains relatively low when compared to the impact of environmental factors. The MiBioGen consortium—a meta-analysis of 18,340 individuals across 24 cohorts—reported 31 genetic loci associated with microbiome traits at genome-wide significance thresholds. Of these, only one—the LCT gene—generated study-wide significance due to its strong association with *Bifidobacterium* abundance [31].

Beyond the LCT gene, several other loci have been linked to microbial composition. For example, variants in FUT2, which encodes a fucosyltransferase involved in mucosal glycan secretion, are associated with differences in *Ruminococcus torques* and *Ruminococcus gnavus* abundance [31]. Similarly, immune-related genes such as CLEC4F and CLEC4A have been connected to microbial metabolic functions related to riboflavin biosynthesis [21]. Additionally, the immune-regulatory genes NOD2 and CARD9 are associated with shifts in *Enterobacteriaceae* populations, particularly in individuals predisposed to inflammatory bowel disease (IBD) [21].

While these findings highlight the role of host genetics in shaping microbiota, overall heritability remains limited. Twin studies suggest that only about 19 bacterial taxa exhibit significant heritability, with most showing a weak to moderate genetic influence [31]. Furthermore, microbiome-wide heritability analyses indicate that population-specific factors—such as diet, geography, and medication use—often outweigh genetic predispositions [31].

Nevertheless, the influence of host genetics on the microbiome function is an emerging area of interest. Some genetic variants may affect not only microbial composition but also metabolic pathways. For instance, GWASs have identified host variants influencing the metabolism of BAs, which, in turn, select specific microbial taxa [21]. Genes involved in nutrient metabolism, such as SORCS2 and SLIT3, have also been associated with microbial pathways related to plant sterol degradation and obesity-linked metabolic activity [21].

In summary, while genetic variation contributes to inter-individual differences in microbiota composition and function, environmental and lifestyle factors play a far more dominant role. These findings underscore the importance of considering both genetic and external influences in the design of personalized microbiome-based interventions [32].

## 3. The Gut Microbiome as an Endocrine Modulator of Reproductive Signaling

The gut microbiota plays a critical role in host metabolism, immune function, and endocrine regulation. It is now widely recognized as a virtual endocrine organ due to its ability to produce and regulate hormonal signals that influence distant organs. One of its key endocrine functions is its involvement in estrogen metabolism, primarily through the activity of the estrobolome [33]. The interaction between the gut microbiome and hormonal balance has significant implications for reproductive health, metabolic disorders, and hormone-dependent conditions such as PCOS and endometriosis [2].

### 3.1. Estrobolome-Driven Estrogen Recycling: Molecular Links to Endometrial Receptivity

Estrogen metabolism involves a three-phase process comprising hepatic conjugation, microbial deconjugation, and excretion [34].

In the liver, estrogen is conjugated into water-soluble metabolites—such as estrone sulfate and estradiol glucuronide—facilitating their biliary excretion into the gastrointestinal tract. However, the gut microbiota plays a pivotal role in determining whether such conjugated estrogen is reabsorbed or eliminated [34].

Specific gut bacteria—including *Clostridium*, *Escherichia*, *Bacteroides*, and *Lactobacillus*—produce β-glucuronidase, an enzyme that deconjugates estrogen metabolites, enabling their reabsorption into the systemic circulation. This process, known as enterohepatic recirculation, regulates estrogen bioavailability and contributes to hormonal homeostasis [33].

Dysbiosis, or a disruption in gut microbial balance, can markedly alter estrogen metabolism. A reduction in β-glucuronidase-producing microbes may impair estrogen reabsorption, leading to systemic estrogen deficiency and negatively affecting reproductive and metabolic function. Conversely, overrepresentation of these bacteria can result in excessive estrogen recirculation, which has been associated with estrogen-dependent conditions such as breast cancer, endometriosis, and infertility [35,36,37].

### 3.2. Microbial Modulation of Metabolic Hormones: Downstream Effects on Implantation and Uterine Function

The gut microbiota shapes the secretion and function of several key metabolic hormones via microbial metabolites, particularly SCFAs such as butyrate and propionate. These byproducts regulate enteroendocrine cell signaling, hormone release, and glucose and energy metabolism, reinforcing the microbiota’s emerging role as a metabolically active endocrine organ [38].

As summarized in Table 1, SCFAs and other microbial-derived metabolites modulate a wide range of hormones, including glucagon-like peptide-1 (GLP-1), peptide YY (PYY), ghrelin, leptin, insulin, and central neuropeptides such as neuropeptide Y (NPY) and orexin [39]. Collectively, these hormones regulate appetite, satiety, insulin sensitivity, lipid homeostasis, and neuroendocrine function [40]. The microbial modulation of these signaling pathways presents promising therapeutic opportunities through dietary, probiotic, and prebiotic interventions designed to improve both metabolic and reproductive outcomes.

## 4. Microbiota-Mediated Mechanisms Underlying Endometrial Dysfunction in Gynecological Disorders

The gut microbiota plays a central role in maintaining systemic immune balance, hormonal regulation, and metabolic homeostasis—all of which are fundamental to reproductive health. Growing evidence indicates that gut dysbiosis contributes to the pathogenesis of various gynecological disorders, including endometriosis, PCOS, RPL, and gynecologic cancers. These conditions often share common underlying mechanisms, such as chronic inflammation, immune dysregulation, and hormonal imbalance—frequently mediated by the gut–endometrial microbiota axis [1,41]. The key regulatory pathways influenced by microbiota are summarized in Table 2.

### 4.1. Endometriosis Pathogenesis Through Gut-Driven Inflammatory and Estrogenic Dysregulation

Endometriosis is an estrogen-dependent inflammatory condition that affects approximately 6–10% of reproductive-age women. It is characterized by the ectopic growth of endometrial-like tissue outside the uterus. The gut microbiota is increasingly recognized as a key factor in the pathogenesis of endometriosis through its influence on immune regulation, estrogen metabolism, and inflammatory signaling pathways. Both experimental studies and clinical trials now provide compelling evidence that gut microbial composition and function significantly contribute to the development and severity of endometriotic lesions.

Elevated levels of Escherichia coli and Bacteroides in the gut microbiota have been linked to increased β-glucuronidase activity, contributing to a hormonal milieu that favors endometriosis progression. Recent human studies have demonstrated that elevated β-glucuronidase correlates with the enhanced proliferation and migration of endometrial stromal cells [49,50], increased lesion volume, and impaired macrophage function—further implicating gut dysbiosis in disease exacerbation. Notably, an overrepresentation of M2-like macrophages in the peritoneal cavity of endometriosis patients is associated with persistent inflammation and lesion survival [51,52]. Additionally, gut-derived LPS from Gram-negative bacteria such as E. coli activates Toll-like receptor 4 (TLR4), triggering the increased production of pro-inflammatory cytokines including TNF-α, IL-6, and IL-1β—all of which are elevated in the peritoneal fluid of individuals with endometriosis [53,54,55,56].

Emerging evidence indicates that SCFAs exert protective effects against endometriosis by modulating immune cell function, dampening inflammation, and restoring intestinal barrier integrity. SCFAs—particularly butyrate—promote anti-inflammatory M2 macrophage polarization while suppressing pro-inflammatory M1 phenotypes, thereby reducing TNF-α and IL-6 secretion and limiting immune cell infiltration at the lesion sites. In murine models, butyrate administration leads to significantly smaller endometriotic lesions compared to controls [57,58]. Moreover, SCFAs such as butyrate and propionate enhance regulatory T cell (Treg) differentiation by activating G-protein-coupled receptors (GPR41, GPR43, and GPR109A) and inhibiting histone deacetylases (HDACs), promoting the expansion of Foxp3^+^ Tregs [59]. This immunological shift toward tolerance downregulates pro-inflammatory cytokines and limits lesion persistence [60].

Among the altered fecal metabolites identified in endometriosis, quinic acid has emerged as a key compound promoting lesion growth [61]. Recent research underscores the intricate role of gut microbiota-derived tryptophan metabolites in the pathogenesis of endometriosis, revealing a complex interplay between microbial metabolism, immune modulation, and inflammation. The kynurenine pathway (KP)—the dominant route of tryptophan metabolism—is upregulated in endometriosis patients, resulting in elevated kynurenine levels that suppress effector T cell activity and promote regulatory T cell (Treg) expansion, thereby facilitating immune tolerance toward ectopic lesions [62]. Experimental studies have demonstrated that the inhibition of indoleamine 2,3-dioxygenase (IDO-1)—the enzyme catalyzing the first step of the KP—reduces lesion size and reverses local immunosuppression, highlighting a potential therapeutic target [62].

Conversely, indolepropionic acid (IPA)—a beneficial microbial metabolite produced via the indole pathway—exerts anti-inflammatory and antioxidant effects by activating the aryl hydrocarbon receptor (AhR). This activation suppresses IL-6 and TNF-α, which are two pro-inflammatory cytokines prominently involved in endometriosis pathophysiology [63]. Fecal metabolomic studies have shown significantly reduced levels of IPA in endometriosis patients [64], correlating with increased systemic inflammation and lesion proliferation. Other indole derivatives, including indole-3-acetate (IAA) and indole-3-lactic acid (ILA), further shape immune responses by enhancing IL-22 production and promoting Treg expansion—mechanisms that may contribute to fibrosis and lesion maintenance in advanced disease. Collectively, these findings suggest that interventions targeting the tryptophan–AhR axis—such as probiotics, prebiotics, IDO-1 inhibitors, and AhR modulators—represent a promising, non-hormonal approach to managing endometriosis through microbiome-based immunomodulation [65].

Beyond metabolomic dysregulation, recent evidence implicates Fusobacterium infection as a direct microbial driver of endometriosis progression, linking reproductive tract dysbiosis to lesion development. In a recent cohort study, Fusobacterium was detected in the endometrial tissue of 64% of women with endometriosis compared to less than 10% of controls—indicating a strong microbial association with disease pathology [66]. Mechanistic studies revealed that Fusobacterium activates transforming growth factor-β (TGF-β) signaling, inducing the transdifferentiation of endometrial fibroblasts into transgelin (TAGLN)-positive myofibroblasts—a key process underlying lesion fibrosis and survival. In a syngeneic mouse model, intravaginal inoculation with Fusobacterium resulted in significantly larger and more numerous lesions, whereas targeted antibiotic therapy reduced lesion formation and reversed fibroblast activation [66,67].

These findings correlate with prior evidence on tryptophan metabolism: kynurenine-mediated immunosuppression and IPA deficiency jointly promote lesion tolerance, inflammation, and fibrotic remodeling. Together, they suggest a unified model in which gut and reproductive tract dysbiosis cooperatively drives immune evasion, chronic inflammation, and endometriotic lesion progression. Future therapeutic strategies—targeting Fusobacterium, restoring SCFA and IPA production, or inhibiting the kynurenine pathway—may offer innovative approaches for the prevention and treatment of endometriosis.

### 4.2. Gut Microbial Dysbiosis in PCOS: Crosstalk Between Inflammation, Hormonal Imbalance, and Endometrial Disruption

Polycystic ovary syndrome, a complex endocrine disorder and one of the foremost causes of anovulatory infertility has increasingly been associated with gut microbial dysbiosis. This disruption contributes to the pathogenesis of PCOS via intertwined metabolic, inflammatory, and hormonal pathways. Notably, women with PCOS often exhibit a perturbed Firmicutes/Bacteroidetes ratio, which contributes to gut barrier dysfunction and endotoxemia through elevated LPS translocation. The resulting low-grade systemic inflammation exacerbates insulin resistance and hyperandrogenism—hallmark features of PCOS that impair folliculogenesis and ovulatory function [47,48].

Beyond taxonomic imbalance, a reduction in SCFA-producing bacteria—particularly *Faecalibacterium prausnitzii*, *Roseburia* spp., and *Eubacterium rectale*—has been observed, leading to diminished levels of SCFAs, which is a key regulator of intestinal integrity and immune modulation [68]. In an experimentally induced PCOS rat model, supplementation with sodium acetate—a SCFA and histone deacetylase (HDAC) inhibitor—ameliorated hyperinsulinemia, dyslipidemia, and ovarian oxidative stress while restoring 17β-estradiol levels, sex hormone-binding globulin (SHBG) expression, and ovarian histoarchitecture. These effects were mediated via HDAC suppression and the subsequent upregulation of nuclear factor erythroid 2–related factor 2 (Nrf2), enhancing antioxidant defenses (e.g., glutathione peroxidase and GSH) and reducing TNF-α expression [69]. Importantly, acetate-treated PCOS rats exhibited marked improvements in insulin sensitivity and the normalization of the LH/FSH ratio, suggesting the reversal of neuroendocrine disruption. These findings provide compelling in vivo evidence that SCFA supplementation—whether through microbial metabolites or dietary modulation—may offer a promising non-pharmacological strategy for addressing both metabolic and reproductive dysfunctions in PCOS.

Moreover, the dysbiosis-related dysregulation of β-glucuronidase enzyme activity may alter estrogen recirculation, further contributing to the hormonal imbalance observed in PCOS [70]. Gut microbial signals have also been shown to influence the pulsatility of the gonadotropin-releasing hormone (GnRH) via neuroendocrine–immune signaling pathways, suggesting its upstream regulatory role in reproductive hormone dynamics [3]. Notably, animal studies have shown that FMT from PCOS models can recapitulate reproductive and metabolic phenotypes in germ-free mice, including obesity, lipid and glucose metabolic disturbances, insulin resistance, and disrupted ovarian function. These effects appear to result from microbial interference with host glycemic regulation, adipocyte signaling, and insulin homeostasis, ultimately driving pancreatic and ovarian dysfunction. This evidence can suggest a causal relationship between dysbiosis and PCOS pathophysiology, reinforcing the microbiota as both a mediator and therapeutic target [71].

Emerging evidence also supports the bidirectional gut-reproductive axis, where chronic inflammation, microbial metabolites, and immune dysregulation feed into the impaired follicular microenvironment and endometrial signaling. Given this multifaceted interplay, interventions targeting the gut microbiome—via personalized probiotics, high-fiber diets, or FMT—may represent promising adjuncts in PCOS treatment paradigms, warranting further investigation in controlled clinical trials [52,72].

### 4.3. Actions of Gut–Immune–Endometrial Axis in Recurrent Implantation Failure (RIF) and Recurrent Pregnancy Loss (RPL)

Recurrent implantation failure and recurrent pregnancy loss, while clinically distinct, share overlapping etiological and immunological features that justify their consideration within a unified framework. Both conditions are associated with impaired maternal–embryo communication, dysregulated immune tolerance, and suboptimal endometrial receptivity despite the presence of morphologically normal embryos. Increasing evidence implicates systemic inflammation, altered cytokine signaling, and microbiota-derived metabolic cues in the disruption of early pregnancy establishment and maintenance. Given these shared pathophysiological mechanisms—particularly those mediated by the gut–immune–endometrial axis—RIF and RPL are addressed together in this section to highlight their convergence at the interface of reproductive immunology and microbial ecology.

Recurrent implantation failure—typically defined as the failure to achieve clinical pregnancy after the transfer of good-quality embryos across at least two or more in vitro fertilization (IVF) cycles— remains a major challenge in reproductive medicine [73]. While uterine and embryonic factors have long been investigated, mounting evidence now indicates that the gut microbiome is a previously underappreciated modulator of endometrial receptivity through systemic immune and metabolic regulation [73,74].

Although classical etiologies—such as chromosomal abnormalities, uterine anomalies, thrombophilia, and endocrine dysfunction—account for a subset of cases, a growing body of research identifies immune dysregulation and microbiome perturbations as critical contributors to unexplained RPL and RIF [73,74]. Recent findings underscore the pivotal role of the gut microbiome in regulating systemic maternal immunity, modulating endometrial immune cell function, and producing metabolites that promote tolerance—each of which is essential for successful implantation and early fetal development.

High-throughput sequencing and multi-omics analyses have revealed that women with RPL exhibit distinct gut microbial profiles [42], frequently characterized by reduced microbial diversity and the enrichment of pro-inflammatory taxa. To clarify the mechanisms through which gut dysbiosis contributes to pregnancy failure, we delineate four interrelated domains by which the microbiota exerts its influence:(1)The disruption of microbial diversity and ecological stability;(2)The skewing of immune cell polarization and cytokine networks;(3)The dysregulation of microbiota-derived metabolites crucial for immune tolerance;(4)Immunogenetic susceptibility, which is mediated by molecular mimicry and autoimmune activation.

While vaginal microbiota—such as *Lactobacillus crispatus* and *Gardnerella vaginalis*—have been implicated in reproductive disorders, they are not typical members of the gut ecosystem. To prevent the conflation of these distinct microbial environments, Table 3 reflects exclusively gut-specific microbial features. The interplay between gut and vaginal microbiota in reproductive health is undoubtedly important, but it warrants a separate analytical framework given their distinct ecological and functional profiles.

#### 4.3.1. Microbial Diversity Collapse and Taxonomic Shifts in RPL and RIF

Across multiple studies [42,43,44], women with RPL consistently exhibit lower gut microbial alpha diversity, as indicated by reductions in Chao1 and Shannon indices. These changes are accompanied by a depletion of protective commensals such as *Faecalibacterium prausnitzii*, *Bifidobacterium* spp., *Lactobacillus* spp., and *Akkermansia muciniphila*, alongside the enrichment of pro-inflammatory taxa including *Prevotella*, *Bacteroides*, *Eubacterium ruminantium* group, *Enterococcus*, and members of the *Erysipelotrichaceae* family.

These taxonomic shifts are not random but functionally convergent, contributing to compromised gut barrier integrity (“leaky gut”), the translocation of LPS, and the systemic activation of the TLR4–NF-κB signaling axis. Circulating microbial products—including LPS, peptidoglycan, and flagellin—activate monocytes and dendritic cells, leading to elevated levels of pro-inflammatory cytokines such as IL-1β, IL-6, TNF-α, and IL-12, all of which are detrimental to maternal–fetal immune tolerance [43].

A central immunological feature of RPL is the skewing of the T cell compartment away from regulatory T cells (Tregs) toward pro-inflammatory Th1 and Th17 phenotypes [43,74]. The depletion of SCFA-producing genera correlates directly with elevated serum levels of IL-17A, IL-17F, TNF-α, IFN-γ, and IL-2—cytokines known to impair decidualization, trophoblast invasion, and spiral artery remodeling [43].

Intriguingly, the abundance of Prevotella_1 and IL-17A_1 has been negatively correlated with these pro-inflammatory cytokine levels [43,44], while broader microbial richness has shown an inverse association with Th17-related inflammation across multiple clinical situations [43,59,73,80,81,82]. Immune phenotyping by flow cytometry in both peripheral blood and decidual tissues confirms these findings, showing reduced CD4^+^Foxp3^+^ Treg populations and increased IL-17^+^ T cells in RPL patients [75].

In parallel, dysbiosis in the vaginal and endometrial microbiota has been observed in RPL, marked by a reduction in *Lactobacillus crispatus* and overgrowth of *L. iners*, *Gardnerella*, and *Atopobium* [76]. These shifts promote elevated local levels of IL-6 and IL-8 and recruit cytotoxic natural killer (NK) cells and neutrophils, further exacerbating fetal rejection [43,44].

One of the most striking findings by Liu et al. [43] is the identification of gut-derived metabolites as upstream immunomodulators in RPL. Two histidine metabolism byproducts—imidazolepropionic acid and 1,4-methylimidazoleacetic acid—were significantly elevated in RPL patients and correlated positively with IL-17A, TNF-α, and IFN-γ. These metabolites impair intestinal epithelial integrity, promote oxidative stress, and are predictive of miscarriage risk with high accuracy (AUC > 0.91).

At the same time, anti-inflammatory BAs such as hyodeoxycholic acid and isolithocholic acid—known to promote Treg differentiation via FXR and GPBAR1 signaling [45,46]—are depleted in RPL and inversely correlating with pro-inflammatory cytokine expression [52].

SCFAs, including butyrate and acetate, were also markedly reduced. These microbial metabolites are critical for Foxp3^+^ Treg expansion, histone deacetylase (HDAC) inhibition, and the suppression of NLRP3 inflammasome activation. Their depletion—linked to the loss of *Roseburia*, *Ruminococcaceae*, and *Anaerostipes*—exacerbates Th17-driven inflammation and contributes to decidual immune dysregulation [77,83].

#### 4.3.2. Microbiota-Driven Autoimmunity and Molecular Mimicry in Endometrial Rejection

Microbiota–host crosstalk in RIF and RPL may also involve molecular mimicry, particularly in women carrying HLA-DQ2 and HLA-DQ8 haplotypes [78,84]. These alleles are associated with autoimmune conditions such as celiac disease and autoimmune thyroiditis and are frequently linked to altered gut microbial profiles enriched in Firmicutes, Proteobacteria, and *Enterobacteriaceae*. As highlighted by Garmendia and Vomstein, cross-reactivity between microbial antigens and placental peptides may provoke autoantibody formation and complement activation at the maternal–fetal interface [42,44].

This autoimmune axis is supported by findings of increased anti-phospholipid and anti-nuclear antibodies (ANAs) in RPL patients, often occurring alongside a *Prevotella*-dominant gut microbiota—further implicating dysbiosis in the activation of humoral immune pathways [79]. Notably, anticardiolipin IgG is a prevalent autoantibody in women with RIF. Moreover, several non-criteria anti-phospholipid antibodies—namely anti-β2-glycoprotein I IgA (aβ2GPI-IgA), anti-phosphatidylglycerol IgG (aPG-IgG), and aPG-IgM—also show elevated positivity rates in RIF patients, supporting their potential utility as biomarkers for identifying autoimmune risk in women undergoing IVF treatment [85].

### 4.4. Gut Microbial Dysbiosis in Preterm Birth: Immune Activation and Barrier Dysfunction at the Maternal–Fetal Interface

Preterm birth, defined as delivery before 37 weeks of gestation, is a leading global cause of neonatal morbidity and mortality. Although traditionally attributed to local reproductive tract factors—such as cervical insufficiency and ascending infections—emerging evidence suggests that systemic influences, particularly those originating from the gut microbiota, may play a pivotal role in modulating the timing of parturition via immune and metabolic signaling pathways [86].

Dahl et al. (2017) [87], using 16S rRNA sequencing of postpartum fecal samples from 121 Norwegian women, found that mothers who delivered prematurely exhibited significantly lower gut microbial alpha diversity (Shannon index, Phylogenetic Diversity, and Observed OTUs). This reduction was accompanied by a decreased relative abundance of beneficial taxa, including *Bifidobacterium*, *Streptococcus*, and members of the Clostridiales order. These microbes are known to support regulatory T cell (Treg) expansion and suppress NF-κB-driven inflammation via the production of metabolites such as acetate and butyrate. Notably, low maternal gut diversity was independently associated with increased odds of spontaneous PTB, even after adjustment for known confounders—suggesting a mechanistic role for gut-driven systemic inflammation in PTB susceptibility.

Shiozaki et al. (2014) [88], using terminal restriction fragment length polymorphism (T-RFLP) in a smaller Japanese cohort, reported similar reductions in gut *Clostridium* clusters (IV, XIVa, and XVIII) and Bacteroides among women who delivered preterm. Interestingly, this study also noted a higher abundance of *Lactobacillales* in PTB—a finding not corroborated by Dahl et al. (2017) [87], who instead observed reduced *Streptococcus* (a member of *Lactobacillales*) in the PTB group. These discrepancies may reflect methodological differences (T-RFLP vs. 16S sequencing), cohort size, or population-specific microbial baselines. Nevertheless, both studies converge on the key observation that the reduced abundance of immunoregulatory gut taxa may compromise maternal immune tolerance and facilitate the inflammation-driven initiation of labor [89].

Importantly, both studies highlight the fact that gut-derived taxa such as *Bacteroides fragilis* and *Clostridium* spp. are potent inducers of IL-10-secreting Tregs through mechanisms involving polysaccharide A and microbial metabolite signaling. Their depletion may impair systemic immune regulation, tipping the balance toward a pro-inflammatory state conducive to premature uterine activation.

These findings support the broader concept that gut dysbiosis may function as an upstream modulator of reproductive inflammation. Given the anatomical proximity of the rectum to the lower reproductive tract, it is plausible that microbial translocation or seeding could further influence the vaginal milieu, compounding the risk of PTB.

While causality has yet to be definitively established, current evidence supports the existence of a gut–systemic–uterine inflammatory axis involved in birth timing regulation. Future research should prioritize the identification of predictive microbial signatures and metabolomic biomarkers for PTB risk. Additionally, intervention studies are warranted to assess whether strategies that enhance gut microbial diversity—such as probiotic supplementation or prebiotic-rich diets—can prevent adverse birth outcomes.

### 4.5. Gut-Endometrial Crosstalk in Preeclampsia: Microbial Influences on Vascular Inflammation and Placental Signaling

Preeclampsia is a complex hypertensive disorder of pregnancy characterized by new-onset hypertension and proteinuria after 20 weeks of gestation. It poses serious risks to both maternal and fetal health. Recent research has illuminated the pivotal role of the gut microbiome in modulating systemic inflammation, immune responses, and metabolic signaling, all of which are critical in the pathogenesis of PE [90].

Multiple studies have shown that individuals with PE exhibit distinct alterations in gut microbiota composition compared to normotensive pregnant women. Most notably, there is a significant reduction in beneficial SCFA-producing bacteria, including *Akkermansia muciniphila*, *Faecalibacterium prausnitzii*, and *Bifidobacterium* spp. In parallel, there is an overrepresentation of pro-inflammatory taxa from the Proteobacteria and Bacteroidetes phyla. This dysbiotic shift is associated with increased intestinal permeability and elevated levels of circulating LPS, which, in turn, trigger systemic inflammation—a hallmark of PE pathology [91].

The loss of SCFA-producing bacteria results in decreased SCFA availability, which is essential for the maintenance of regulatory T cells (Tregs) and immune tolerance [92]. This imbalance contributes to an inflammatory intrauterine environment detrimental to endometrial receptivity and proper placental development.

In addition to SCFAs, other gut-derived microbial metabolites, such as trimethylamine-N-oxide (TMAO), have been implicated in PE [89,90]. TMAO is produced through the microbial metabolism of dietary choline and carnitine [91], and elevated plasma levels have been reported in PE patients [92]. TMAO is known to promote endothelial dysfunction, oxidative stress, and hypertension, thereby exacerbating the pathophysiological cascade of PE [93,94,95,96,97].

These findings support the concept of a gut–systemic–endometrial axis in PE, wherein dysbiosis and altered microbial metabolites drive systemic immune and vascular dysfunction. Key microbial metabolites and their mechanistic links to PE are summarized in Table 4.

The key host pathways and outcomes in PE include several important signaling mechanisms. The TLR4-NF-κB axis [100], which is activated by LPS, contributes to the exacerbation of systemic inflammation and vascular dysfunction. Additionally, SCFA-GPCR signaling (GPR41/43, GPR109A) [98] plays a role in regulating blood pressure, anti-inflammatory cytokines, and vascular tone. Another significant pathway involves HDAC inhibition by butyrate, which modulates the epigenetic regulation of endothelial and immune functions [92,101]. Finally, bile acid signaling through FXR/GPBAR1, which is typically suppressed in PE [102,103], normally promotes Treg cells and maintains vascular homeostasis.

## 5. Microbial Dysbiosis Beyond Classical Gynecological Disorders: Endocrine–Immune Disruption and Endometrial Signaling

Beyond well-characterized conditions such as endometriosis, PCOS, RIF, and PE, a growing array of gynecologic and reproductive disorders have been increasingly linked to gut microbial dysbiosis. These conditions converge on three key interrelated domains: altered estrobolome activity, immune dysregulation, and chronic inflammation driven by microbial metabolites and cytokines.

Disruptions in the gut microbiota impact not only local mucosal immunity but also systemic immune priming and hormonal homeostasis, collectively triggering cascades that impair reproductive function. This broader perspective highlights the microbiota as a central player in immune–endocrine crosstalk relevant to both gynecological pathology and fertility. A summary of these pathological conditions and their underlying microbiome-mediated mechanisms is provided in Table 5.

### 5.1. Bacterial Vaginosis and the Gut–Vaginal Axis: Microbial Crosstalk and Endometrial Consequences

The gut microbiota plays a critical role in maintaining vaginal microbial stability via systemic immune and metabolic signaling, constituting a functional gut–vaginal axis. In BV, the depletion of protective *Lactobacillus* species enables the overgrowth of anaerobic bacteria such as *Gardnerella*, *Atopobium*, and *Mobiluncus*, promoting local inflammation, biofilm formation, and increased mucosal permeability linking (IBD) [107].

These alterations are associated with elevated levels of pro-inflammatory cytokines, including IL-6, TNF-α, and IL-1β, which contribute to immune dysregulation and increase the risk of vaginosis [108]. Emerging evidence suggests that specific gut microbial taxa can exert causal effects on vaginal health via systemic metabolic mediators. For example, *Candidatus soleaferrea* and *Dialister* have been identified as gut-derived risk factors for vaginitis, with their pathogenic effects attenuated by circulating plasma metabolites—notably 3-phosphoglycerate (in the case of *C. soleaferrea*) and lysophosphatidylcholine (for *Dialister*) [109].

In contrast, certain commensal bacteria may exert protective effects. *Lachnospiraceae* UCG-008 has been associated with reduced vaginitis risk through favorable shifts in metabolic ratios, including an increased arachidonate/pyruvate ratio and a decreased palmitate/myristate ratio—both indicative of anti-inflammatory metabolic states [109]. *Ruminiclostridium* also appears to provide protection against vaginitis, although the specific mediating metabolites remain to be identified.

Notably, SCFAs—widely recognized for their anti-inflammatory and homeostatic roles in the gut—may exhibit context-dependent effects in the vaginal environment, underscoring the importance of site-specific microbiota–host interactions.

While in the gut, SCFAs like butyrate and propionate support epithelial barrier function, immune tolerance, and the suppression of systemic inflammation; elevated vaginal SCFAs have been linked to increased pH, the depletion of Lactobacilli, and pro-inflammatory cytokine production, contributing to a dysbiotic state [110]. This dichotomy highlights the niche specificity of microbial metabolite functions and underscores the need for caution when extrapolating gut-derived benefits to the reproductive tract.

Together, these findings support a metabolically mediated gut–vaginal axis, in which systemic metabolites shaped by gut microbial composition influence vaginal immunity and resilience. Modulating these pathways may offer new avenues for preventing or treating vaginal dysbiosis and its reproductive complications.

### 5.2. Uterine Fibroids: Estrogen Dysregulation and Immune Modulation Mediated by the Gut Microbiota

Emerging evidence suggests that gut microbiota dysbiosis may serve as a risk factor for uterine fibroids or modulate disease progression. Alterations in gut microbial composition—including shifts in the relative abundance of Firmicutes, Proteobacteria, Actinobacteria, and Verrucomicrobia—have been detected in fecal samples of women with fibroids [104].

Uterine fibroids are estrogen-responsive tumors, and their growth has been linked to the heightened activity of the estrobolome, particularly through the increased expression of β-glucuronidase by gut-resident bacteria such as *Clostridia*. This enzyme facilitates the deconjugation and enterohepatic recirculation of estrogens, thereby raising systemic estrogen levels and promoting fibroid proliferation [8,111].

In addition to hormonal influences, fibroid tissue displays infiltration by pro-inflammatory macrophages and elevated levels of interleukin-6 (IL-6) [112,113]. This inflammatory milieu contributes to fibrotic remodeling, aberrant immune surveillance, and disease persistence.

### 5.3. Gynecologic Cancers: Gut Microbiota, Inflammation, and Hormone-Driven Oncogenesis

An increased abundance of pro-inflammatory bacteria in the gut and genital tract—such as *Fusobacterium*, *Bacteroides*, and *Escherichia coli*—has been associated with elevated levels of IL-6, TNF-α, and reactive oxygen species (ROS), driving chronic inflammation and promoting DNA damage, angiogenesis, and immune evasion [114,115]. Conversely, the depletion of *Lactobacillus* spp. compromises mucosal integrity and impairs anti-tumor immune responses, including natural killer (NK) cells and cytotoxic T cell activity, contributing to tumor progression in endometrial and cervical cancers [105,114,115].

Recent Mendelian randomization (MR) analyses suggest a causal link due to correlational findings in the gut microbiota composition and gynecologic cancer risk, offering novel insights into microbial drivers of oncogenesis with direct implications for reproductive health. Kong et al. [105] conducted a two-sample MR study identifying 33 suggestive causal relationships between genetically predicted gut microbial taxa and risks of ovarian, endometrial, and cervical cancers. Among these, 11 microbial genera were associated with increased cancer risk, while 19 showed protective associations.

Notably, the bidirectional design of the study revealed that gynecologic cancers may themselves influence gut microbiota composition, supporting a dynamic and reciprocal gut–reproductive tract axis. These findings underscore that microbiota are not passive bystanders but are active modulators of gynecologic health, acting through immunological, endocrine, and potentially epigenetic mechanisms.

The overrepresentation of pro-inflammatory or estrogen-metabolizing microbes may prime the endometrium and ovarian epithelium for malignant transformation via chronic low-grade inflammation, enhanced β-glucuronidase activity, and disrupted estrogen metabolism. The reverse association—where the presence of cancer alters microbial ecosystems—further reinforces the concept of a bidirectional relationship between the gut and reproductive tract, with important implications for cancer prevention, progression, and fertility outcomes [106].

### 5.4. Gut Dysbiosis-Driven Immune Priming and Hormonal Imbalance in Reproductive Dysfunction

Beyond malignancy, similar immunoendocrine disruptions extend to other gynecologic conditions, including pelvic inflammatory disease (PID) and menstrual irregularities. Though distinct in etiology, these disorders share overlapping microbial signatures and pathophysiological mechanisms—including inflammatory signaling, dysregulated hormone metabolism, and epithelial immune priming.

PID is mostly initiated by ascending genital tract infections, yet gut and vaginal dysbiosis may exacerbate systemic immune activation and perpetuate chronic inflammation. The overgrowth of microbial taxa such as *Gardnerella*, *Mycoplasma*, and *Bacteroides* can induce TLR4-mediated signaling and stimulate the release of pro-inflammatory cytokines, including IL-1β, IL-8, and TNF-α [116]. This sustained inflammatory environment within the pelvis contributes to fallopian tube scarring and elevates the risk of tubal factor infertility [2].

Menstrual irregularities have also been linked to gut dysbiosis, particularly reduced microbial diversity and the enrichment of *Bacteroides* and *Clostridium* species, which are associated with disrupted steroid hormone metabolism [117]. Increased microbial β-glucuronidase activity promotes estrogen recirculation, potentially resulting in luteal phase defects and irregular menstrual cycles. Concurrently, dysbiosis-induced systemic inflammation—characterized by elevated IL-6 and C-reactive protein (CRP)—can impair hypothalamic–pituitary–gonadal (HPG) axis signaling, contributing to anovulation and cycle irregularities [47,48].

## 6. Gut Microbiota and Endometrial Biology in Reproductive Function: Mechanistic Insights and Future Perspectives

Gut dysbiosis contributes to at least six major categories of endometrial dysfunction:(1)Immune dysregulation, characterized by elevated IL-6, TNF-α, and IL-1β; increased Th17 cells; and reduced Tregs [118];(2)The activation of inflammatory signaling pathways, including NF-κB, TLR4–LPS, and NLRP3 inflammasome [118];(3)The downregulation of receptivity markers such as LIF, HOXA10, integrin αvβ3, MUC1, and glycodelin [119,120];(4)Hormonal imbalance, primarily driven by elevated microbial β-glucuronidase activity, resulting in estrogen dominance, progesterone resistance, and ERβ upregulation [33,121];(5)Epithelial barrier dysfunction through reduced tight junction proteins and increased matrix metalloproteinases (MMPs) [122,123]; and(6)Maladaptive microbial–endometrial crosstalk involving translocated microbial metabolites, local immune activation, and shifts in the endometrial microbial community [74].

These dysbiotic pathways have been implicated in a wide range of reproductive and gynecological disorders, including not only classical implantation-related failures—such as recurrent implantation failure (RIF), recurrent pregnancy loss (RPL), chronic endometritis, and endometriosis—but also systemic or hormonally driven conditions such as polycystic ovary syndrome (PCOS), preeclampsia (PE), and preterm birth (PTB). Additionally, uterine fibroids, bacterial vaginosis (BV), pelvic inflammatory disease (PID), menstrual irregularities, and even gynecologic cancers (e.g., endometrial and cervical) have shown overlapping microbial and immunoendocrine features that point toward a common upstream influence—gut microbial imbalance.

An emerging area of interest is the translocation of microbial DNA or fragments from the gut to the peritoneal cavity [124], particularly in the context of increased intestinal permeability and mucosal barrier dysfunction. These microbial elements may reach the peritoneal environment or even the endometrium, where they interact with local immune cells via pattern recognition receptors (e.g., TLR9), triggering pro-inflammatory cascades that impair implantation. The detection of microbial DNA in the peritoneal fluid of women without clinical infection strongly supports the concept of a gut–peritoneal–endometrial axis. We propose that such microbial translocation contributes to the parainflammation and immune priming of the endometrial microenvironment [125].

In addition to cell-free microbial components, recent evidence highlights the possibility of the direct translocation of viable microbes across mucosal and systemic compartments. Takada et al. (2023) [126] describe proximal anatomical trafficking between the rectum and vagina and the hematogenous dissemination of bacteria from the gut and oral cavity to the uterus under inflammatory or barrier-compromised conditions. Similarly, Łaniewski et al. (2020) [127] report the phylogenetic overlap of species such as *Lactobacillus crispatus* and *Gardnerella vaginalis* in both bladder and vaginal samples of the same individuals—reinforcing the concept of viable microbe sharing across urogenital niches. These findings extend the paradigm of microbial translocation beyond inert fragments, suggesting that live, potentially immunomodulatory organisms may influence local immunity, endometrial signaling, and implantation potential.

Beyond implantation failure, microbial translocation and dysbiosis-related inflammation may also contribute to the fibrogenic remodeling of the reproductive tract. Recent research highlights the crosstalk between gut, circulatory, and peritoneal microbiota in the activation of fibrotic signaling pathways. Microbial metabolites such as LPS, indoxyl sulfate (IS), and p-cresyl sulfate (pCS) have been shown to activate TGF-β/Smad, NF-κB, and mTOR pathways—leading to mesothelial or epithelial–mesenchymal transitions (MMT/EMT) and extracellular matrix deposition [128]. While initially characterized in the context of peritoneal dialysis and pulmonary fibrosis, these mechanisms are now hypothesized to underline pathological uterine remodeling in conditions such as adenomyosis, Asherman’s syndrome, and fibrotic subtypes of endometriosis.

A central mechanism by which the gut microbiota shapes reproductive outcomes lies in its regulation of estrogen metabolism. Through the activity of the estrobolome, specific gut bacteria—particularly those producing β-glucuronidase such as *Clostridia*—can deconjugate estrogen in the intestine, thereby enhancing enterohepatic recirculation and increasing systemic estrogen levels [2]. This mechanism plays a key role in estrogen-sensitive disorders, including endometriosis, uterine fibroids, and gynecologic cancers such as endometrial and cervical cancer. Dysbiosis-induced alterations in estrogen homeostasis may also impair the tightly regulated hormonal signaling required for endometrial proliferation, decidualization, and implantation.

Beyond estrogen metabolism, the gut microbiota exerts broader systemic metabolic control. It modulates insulin sensitivity, leptin signaling, and lipid homeostasis—all of which intersect with ovulatory function and endometrial transformation. These mechanisms are especially relevant in PCOS, where insulin resistance and hyperandrogenism contribute to both metabolic and reproductive dysfunction. Similarly, altered gut microbial profiles have been associated with PE, gestational hypertension, and dyslipidemia, further highlighting the endocrine significance of microbial networks [45,48].

The gut microbiota also participates in the regulation of menstrual cyclicity. Increased microbial β-glucuronidase activity, when unopposed, may lead to estrogen dominance and luteal phase defects, contributing to irregular menses and menstrual disorders. Concurrently, low-grade inflammation driven by microbial metabolites such as LPS and reactive oxygen species (ROS) can impair hypothalamic–pituitary–gonadal (HPG) axis signaling, further exacerbating ovulatory and menstrual irregularities, as observed in PCOS and chronic anovulation.

In addition to hormonal and metabolic regulation, the gut microbiota plays a pivotal role in shaping the immune environment required for uterine receptivity. A tolerogenic immune milieu—characterized by elevated regulatory T cells (Tregs), balanced Th1/Th2 responses, and controlled pro-inflammatory cytokine activity—is essential for embryo implantation and early pregnancy maintenance. Dysbiosis skews this balance toward Th17 polarization and the increased production of cytokines such as IL-6, IL-17A, and TNF-α, which impair endometrial receptivity and tolerance [129,130]. This mechanism is increasingly recognized in RPL and RIF.

Among the most functionally significant microbial products are SCFAs—notably butyrate and propionate—which influence Treg differentiation, suppress NF-κB signaling, and maintain epithelial integrity [108]. These metabolites have been directly linked to immunotolerance [5] and may critically affect implantation success. SCFA-producing taxa such as *Faecalibacterium prausnitzii*, *Roseburia* spp., and *Bifidobacterium* are frequently depleted in women with RPL, chronic endometritis, and PE [23,57,58,61,92], suggesting a microbial contribution to the loss of pregnancy-related immune control.

Tryptophan metabolism provides another key immunomodulatory pathway. Gut microbiota converts tryptophan into kynurenine and indole derivatives such as indolepropionic acid (IPA). Kynurenine, acting through the aryl hydrocarbon receptor (AhR), promotes Treg expansion and mucosal tolerance. IPA, a potent antioxidant, reinforces epithelial barrier integrity and suppresses local inflammation [6]. Perturbations in these pathways have been linked to RIF, PTB, and RPL [42,83], where tryptophan depletion and altered microbial catabolism may impair both mucosal and systemic immune control [6].

To visually synthesize these complex interactions, Figure 1 presents a schematic model depicting the differential impact of gut microbiota composition—eubiotic versus dysbiotic—on endometrial molecular signaling, immune regulation, and implantation potential.

In a eubiotic state, dominant taxa such as *Faecalibacterium prausnitzii*, *Roseburia* spp., *Bifidobacterium* spp., and *Lactobacillus* spp. contribute to a protective immunoendocrine environment through the production of SCFAs, tryptophan metabolites (e.g., kynurenine, IPA), and anti-inflammatory cytokine profiles [6]. These microbial products foster epithelial integrity, promote Treg expansion, and maintain estrogen balance, all of which are conducive to successful implantation.

In contrast, a dysbiotic gut microbiome—marked by the overgrowth of pathobionts such as *Escherichia coli*, *Bacteroides fragilis*, and pathogenic *Clostridium* species—produces harmful metabolites including LPS and trimethylamine-N-oxide (TMAO) [48]. These compounds activate NF-κB, TLR4, and NLRP3 inflammasome pathways [30], promoting systemic and endometrial inflammation, increasing oxidative stress, and impairing endometrial receptivity. Simultaneously, elevated β-glucuronidase activity enhances estrogen recirculation, contributing to estrogen dominance and progesterone resistance, which are features commonly observed in PCOS, fibroids, and endometriosis [4,77].

As illustrated in the figure, this multifactorial cascade links gut dysbiosis to a broad spectrum of reproductive disorders, including RIF, RPL, PE, BV, PTB, adenomyosis, and gynecologic malignancies. This schematic underscores the central role of the gut microbiota in regulating not just endometrial function but the broader immunoendocrine axis that governs reproductive health and fertility [2,9].

Taken together, these findings support a unifying model in which gut dysbiosis acts as a central upstream driver of diverse reproductive pathologies, linking metabolic, immune, and hormonal dysfunction to impaired endometrial receptivity and implantation failure. The gut–endometrial axis is increasingly recognized as a dynamic interface, mediating both systemic signals and localized immune priming that shapes reproductive outcomes.

Looking ahead, future research should prioritize the following areas:(1)The mechanistic dissection of SCFA- and tryptophan-mediated gene regulation;(2)An investigation into the inflammatory and fibrotic consequences of microbial DNA and viable microbe translocation;(3)The identification of reproducible microbial and metabolic biomarkers across fertility-relevant conditions;(4)The development of precision microbiota-based therapies tailored to specific immunoendocrine profiles.

Advances in multi-omics technologies—including metatranscriptomics, culturomics, metabolomics, and spatial microbiology—will be critical to resolving the temporal and functional relationships between gut dysbiosis and reproductive pathology. The integration of microbial ecology, molecular immunology, and reproductive medicine promises to not only transform our understanding of infertility but also to redefine diagnostic and therapeutic strategies.

Ultimately, the gut–endometrial interface may serve as both a diagnostic window and a therapeutic target, ushering in a new era of personalized fertility medicine, where the modulation of the microbiome is as foundational as hormonal or surgical interventions.

## 7. Conclusions: Integrating Microbial, Immune, and Hormonal Networks in Fertility Research

Female reproductive and gynecological health depends on a delicate balance of metabolic, hormonal, immune, molecular, and microbial systems. The gut microbiota, a critical sensor and modulator of systemic physiology, is pivotal in sustaining this harmony. Disruptions to this balance—triggered by dietary imbalances, environmental exposures, systemic inflammation, or microbial dysbiosis—can initiate a cascade of interdependent feedback loops. These dysregulated processes often amplify each other, driving cycles of dysbiosis, immune dysfunction, hormonal dysregulation, and fibrotic remodeling, which, in turn, precipitate implantation failure, infertility, or chronic reproductive disorders.

From this perspective, effective therapies must address these complex interactions holistically. As reproductive medicine shifts toward precision-based approaches, restoring fertility relies on re-establishing functional harmony across physiological networks. This entails restoring host–microbiota communication, balancing inflammation and tissue regeneration, and recalibrating immune tolerance and activation.

Although emerging clinical and translational studies increasingly link gut dysbiosis to adverse reproductive outcomes, much of the evidence remains correlative. Future research must prioritize longitudinal, mechanistic, and interventional studies to clarify causal relationships and unlock the therapeutic potential of the gut–endometrial axis in female reproductive and gynecological health.

## Figures and Tables

**Figure 1 life-15-00762-f001:**
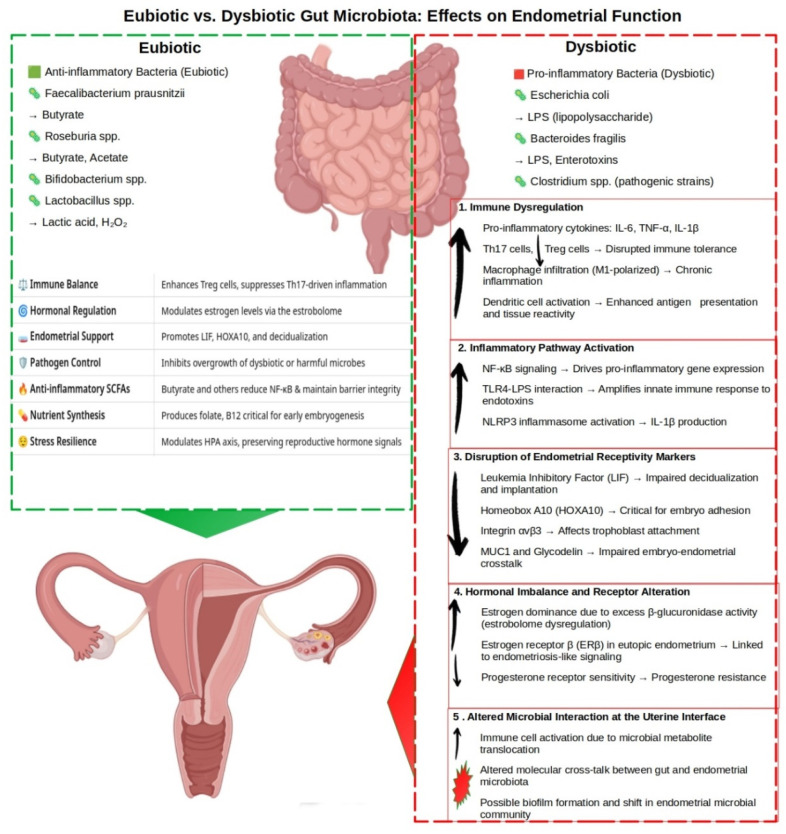
The impact of gut microbiota composition on endometrial function and fertility. This schematic illustrates the differential effects of eubiotic and dysbiotic gut microbiota on endometrial molecular and immunological homeostasis. Eubiotic taxa—including *Faecalibacterium prausnitzii*, *Roseburia* spp., *Bifidobacterium* spp., and *Lactobacillus* spp.—produce short-chain fatty acids (SCFAs) such as butyrate, acetate, and lactic acid, which support the anti-inflammatory, immune tone, maintain epithelial integrity, and enhance endometrial receptivity. In contrast, dysbiotic taxa—such as *Escherichia coli*, *Bacteroides fragilis*, and pathogenic *Clostridium* spp.—generate pro-inflammatory metabolites, including LPS, enterotoxins, and trimethylamine-N-oxide (TMAO), which promote systemic and endometrial inflammation and impair reproductive function. Arrows indicate changes in the abundance, concentration, or activity of microbial, molecular, or cellular factors within regulatory pathways relative to baseline or dysbiotic states. Upward arrows (↑) denote increases; downward arrows (↓) indicate decreases.

**Table 1 life-15-00762-t001:** **The microbial modulation of metabolic hormones involved in energy homeostasis and potential therapeutic implications**. This table summarizes the key metabolic hormones—including glucagon-like peptide-1 (GLP-1), peptide YY (PYY), ghrelin, leptin, insulin, neuropeptide Y (NPY), and orexin—that regulate appetite, glucose metabolism, satiety, and systemic energy balance. It highlights their physiological roles, modulation by gut microbiota (particularly via short-chain fatty acids, bile acids, and microbial-brain signaling), and the emerging therapeutic potential of microbiome-targeted strategies. These include dietary, probiotic, and prebiotic interventions aimed at optimizing metabolic and reproductive health through endocrine modulation. Adapted from Ashraf and Hassan, 2024 [40].

Hormone	Primary Source	Microbial Influence	Physiological Function	Therapeutic Potential
GLP-1	Intestinal L-cells	SCFAs (butyrate and propionate) upregulate secretion via GPR41/GPR43	Enhances insulin secretion; promotes satiety	Prebiotics and SCFA-promoting probiotics enhance GLP-1 for glycemic control and satiety
PYY	Intestinal L-cells	Stimulated by SCFAs; influenced by microbial density	Inhibits appetite; slows gastric emptying	Prebiotic fibers modulate PYY via SCFA production to reduce appetite and improve weight management
Ghrelin	Stomach X/A-like cells (in fundus)	Modulated by gut microbiota composition; lower in SCFA-rich profiles	Stimulates appetite; regulates energy balance	Microbial modulation (e.g., Akkermansia) may suppress ghrelin to reduce food intake and aid obesity management
Leptin	Adipocytes	Indirectly influenced via gut barrier integrity and systemic signals	Regulates satiety and energy expenditure	Gut barrier restoration via probiotics may enhance leptin sensitivity and reduce inflammation-associated resistance
Insulin	Pancreatic β-cells	Enhanced via SCFA-mediated GLP-1 secretion; impacted by bile acids	Regulates blood glucose levels	SCFA-driven incretin release and bile acid signaling can improve insulin sensitivity and glucose regulation
Neuropeptide Y (NPY)	Hypothalamic arcuate nucleus neurons. (AgRP/NPY neurons)	Suppressed by SCFA signaling and the microbial modulation of CNS	Promotes feeding behavior and energy storage	Diet and microbiota interventions targeting SCFA-NPY pathways may help control appetite and metabolic diseases
Orexin	Lateral hypothalamic area (LHA neurons)	Indirect modulation through microbiota–brain axis	Regulates arousal, wakefulness, and appetite	The probiotic modulation of the gut–brain axis may regulate orexin and improve sleep, mood, and appetite patterns

**Table 2 life-15-00762-t002:** **The microbiota-driven immunological and molecular pathways modulating maternal endometrial function and pregnancy outcomes**. This table summarizes the key immunological and molecular pathways through which maternal gut microbiota influence pregnancy outcomes. Each pathway includes specific microbial taxa (drivers), the relevant microbial or host-derived mediators (e.g., SCFAs, LPS, and cytokines), their cellular or molecular targets within the maternal endometrium or immune system, and their functional consequences on implantation, immune tolerance, placentation, and pregnancy maintenance. All listed microbes are gut-derived, and outcome measures refer to maternal physiological states known to support or impair early gestation.

Regulatory Pathways	Microbial Drivers	Molecular Mediators	Target System	Action on Maternal Immune Interface	Key References
**Immune Homeostasis**	*Faecalibacterium prausnitzii*, *Roseburia*, *Bifidobacterium* spp.	SCFAs (butyrate, acetate), IL-10, TGF-β	CD4^+^ T cells (↑ Treg, ↓ Th17), NK cells	Promotes immune tolerance; suppresses pro-inflammatory cytokines	[42,43]
**Inflammatory Regulation**	Dysbiotic expansion: *Prevotella*, *Erysipelotrichaceae*, *Enterococcus*	LPS, IL-6, IL-1β, TNF-α, IL-17A	TLR4/NF-κB signaling in endometrium and periphery	Induces Th17-skewed inflammation; trophoblast apoptosis	[44,45]
**Metabolite Signaling**	↓ *Akkermansia*, *Anaerostipes*, *Ruminococcaceae*	↓ Bile acids (HDCA, LCA), ↑ Imidazolepropionic acid	FXR/GPBAR1, oxidative stress; epithelial integrity	Disrupted mucosal tolerance; elevated cytokines	[43,46]
**Barrier Function**	*Lactobacillus* spp., *Akkermansia muciniphila*	Mucin; tight junction proteins (ZO-1, claudins)	Intestinal and uterine epithelia	Reduced LPS translocation; protects from “leaky gut”	[42,47]
**Endocrine Modulation**	*Clostridium scindens*, *Bacteroides* spp.	Estrobolome (β-glucuronidase), steroid-modulating enzymes	Estrogen/progesterone bioavailability	Impacts endometrial receptivity; decidualization	[2,48]
**Autoimmune Susceptibility**	Dysbiotic networks in HLA-DQ2/DQ8+ hosts	Molecular mimicry, cross-reactive antigens	Autoantibodies, complement system	Increased maternal immune rejection of fetal cells	[42,44]

Arrows indicate changes in the abundance, concentration, or activity of microbial, molecular, or cellular factors within regulatory pathways relative to baseline or dysbiotic states. Upward arrows (↑) denote increases; downward arrows (↓) indicate decreases.

**Table 3 life-15-00762-t003:** **The mechanistic domains linking gut dysbiosis to reproductive dysfunction**. This table summarizes the four major domains through which alterations in the gut microbial composition and function influence female reproductive health. Each row describes characteristic changes in microbial taxa, the corresponding molecular mediators (e.g., LPS, SCFAs, and autoantibodies), and the downstream immunological or metabolic consequences relevant to implantation, pregnancy maintenance, and immune tolerance. The microbial taxa listed are specific to the gut environment; vaginal or urogenital microbiota are not included. Abbreviations: SCFA, short-chain fatty acids; Treg, regulatory T cells; Th17, T-helper 17 cells; TLR, Toll-like receptor; NF-κB, nuclear factor kappa-light-chain-enhancer of activated B cells; LPS, lipopolysaccharide; IL, interleukin; TNF-α, tumor necrosis factor-alpha; IFN-γ, interferon-gamma; HDCA, hyodeoxycholic acid; isoLCA, isolithocholic acid; ANA, antinuclear antibodies; aPL, anti-phospholipid antibodies.

Mechanistic Domain	Microbial Features	Key Mediators	Immunological/Metabolic Consequences	References
**1. Microbial Diversity Collapse**	↓ *Faecalibacterium*, *Bifidobacterium*, *Akkermansia*, SCFA-producing *Lactobacillus* spp. ↑ *Prevotella*, *Bacteroides*, *Enterococcus*, *Erysipelotrichaceae*	LPS, peptidoglycan, flagellin TLR4 → NF-κB → IL-1β, IL-6, TNF-α	Increased gut permeability; systemic inflammation; impaired maternal–fetal tolerance	[42,43,44]
**2. Th17/Treg Imbalance**	↓ SCFA-producers (*Roseburia*, *Anaerostipes*, *Blautia*) ↑ *Prevotella*, *Escherichia/Shigella*	↓ Tregs, ↑ Th17 IL-17A, IL-6, IL-8, IFN-γ	Impaired decidualization; abnormal NK cell recruitment; pro-inflammatory uterine environment	[43,75,76]
**3. Metabolite Dysregulation**	↓ Butyrate and bile acid-producing genera (*Eubacterium*, *Clostridium XIVa*) ↑ Histidine-metabolizing bacteria	↓ SCFAs, ↓ HDCA and isoLCA ↑ Imidazolepropionic acid	Oxidative stress; barrier dysfunction; altered IL-10/IL-17 signaling	[43,45,52,77]
**4. Immunogenetic Susceptibility**	HLA-DQ2/DQ8+genotype ↑ Firmicutes, Proteobacteria, *Prevotella*	Autoantibodies (ANA, aPL), complement activation	Cross-reactivity to fetal antigens; immune rejection; recurrent miscarriage	[42,44,78,79]

Arrows indicate changes in the abundance, concentration, or activity of microbial, molecular, or cellular factors within regulatory pathways relative to baseline or dysbiotic states. Upward arrows (↑) denote increases; downward arrows (↓) indicate decreases.

**Table 4 life-15-00762-t004:** **The gut microbiota-derived metabolites and pathways implicated in preeclampsia (PE)**. This table summarizes the key gut microbial taxa altered in preeclampsia, their associated metabolites, and the host signaling pathways they influence. The proposed mechanisms link microbial shifts—particularly reduced short-chain fatty acid production and increased endotoxin load—to endothelial dysfunction, immune activation, and systemic inflammation, which are all hallmarks of PE pathophysiology. References support associations between specific taxa, metabolite pathways, and clinical or mechanistic findings relevant to PE development.

Microbial Taxa	Abundance in Pe	Associated Metabolites	Host Pathways Affected	Proposed Mechanism in Pe Pathophysiology	Key References
***Escherichia/Shigella* (Proteobacteria)**	↑ Increased	LPS (endotoxin)	TLR4 → NF-κB → IL-6, TNF-α	Promotes systemic inflammation; endothelial dysfunction	[42,98]
***Blautia* (Firmicutes)**	↓ Decreased	Butyrate and valerate	GPCR41/43 signaling; HDAC inhibition	Anti-inflammatory, vasodilatory; protects endothelial barrier	[98]
** *Eubacterium hallii* **	↓ Decreased	Butyrate	SCFA receptor activation; mitochondrial support	Improves vascular tone; reduces oxidative stress	[98]
***Bifidobacterium* spp.**	↓ Decreased	Acetate and lactate	Enhances mucosal barrier integrity; immune modulation	Loss may increase gut permeability and LPS leakage	[43,98]
** *Subdoligranulum* **	↓ Decreased	Butyrate	Treg induction; anti-inflammatory cytokines	Supports immune tolerance; depletion linked to Th17 shift	[98]
** *Enterobacter* **	↑ Increased	LPS and TMA precursors	TLR4 activation; endothelial stress	Associated with hypertension and cytokine elevation	[98]
** *Akkermansia muciniphila* **	↓ Decreased	Mucin-degradation products	Mucin layer maintenance; gut barrier protection	Depletion leads to “leaky gut” and metabolic inflammation	[42,99]
**General SCFA Producers (*Roseburia*, *Faecalibacterium*)**	↓ Decreased	Butyrate, acetate, and propionate	GPR109A; Treg expansion	Critical for immune balance and endothelial protection	[43,46]

**Table 5 life-15-00762-t005:** **Dysbiosis-associated gynecologic conditions: immune and hormonal mechanisms.** This table summarizes the key reproductive and gynecologic disorders associated with alterations in gut and vaginal microbiota. For each condition, the dominant microbial changes are listed alongside key immune mediators—such as pro-inflammatory cytokines and immune cell shifts—and the resulting pathophysiological effects. The interplay between microbial dysbiosis, immune signaling (e.g., IL-6, TNF-α, CRP), and hormonal dysregulation is highlighted as a shared mechanism underlying inflammation, impaired fertility, and disease progression. References provide supporting evidence for microbiota-driven contributions to each pathology.

Condition	Microbial Features	Key Immune Mediators	Pathophysiological Impact	References
**Bacterial Vaginosis (BV)**	↓ *Lactobacillus*, ↑ *anaerobes* (Gardnerella, etc.)	IL-6, IL-1β, TNF-α	Vaginal inflammation, biofilm formation, ↑ PID, miscarriage, preterm birth risk	[30]
**Uterine Fibroids**	↑ β-glucuronidase (*Clostridia*) → ↑ estrogen	IL-6, M1 macrophages	Estrogen-driven fibroid growth, immune cell infiltration, fibrosis	[104]
**Gynecologic Cancers**	↑ *Fusobacterium*, *Bacteroides*, *E. coli*	IL-6, TNF-α, ROS, ↓ NK cells	Chronic inflammation, immune evasion, mucosal breakdown, tumor promotion	[105,106]
**Pelvic Inflammatory Disease (PID)**	↑ *Gardnerella*, *Mycoplasma*, *Bacteroides*	IL-1β, IL-8, TNF-α	Persistent pelvic inflammation, tubal scarring, infertility	[30]
**Menstrual Irregularities**	↓ diversity, ↑ *Bacteroides*, *Clostridium*	IL-6, CRP, β-glucuronidase	Hormonal imbalance, anovulation, cycle disruption	[8]

## Data Availability

Not applicable.

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
