# Peer review of "The Role of the Gut Microbiota in Female Reproductive and Gynecological Health: Insights into Endometrial Signaling Pathways"

_life, 2025, doi:10.3390/life15050762_

Round 1

Reviewer 1 Report

Comments and Suggestions for Authors

In this review article, Mora and colleagues discuss the interactions between the gut and reproductive organs, with a focus on endometrial molecular biology. Overall, the reviewer found this article interesting and helpful in deepening insights into regulation along the gut-endometrial axis. Below are some suggestions to improve clarity and accuracy.

Major comments;

  1. Please distinguish clearly the gut microbiota and the vaginal microbiota. For example, the authors discuss the role of Lactobacillus crispatus in the gut. However, although crispatus can be detected in the gut, L. crispatus is not the major Lactobacillus, unlike in the vagina. Therefore, the reviewer thinks the role of L. crispatus in the gut is very limited. In this sense, it is inappropriate to cite L. crispatus as a representative example of eubiotic Lactobacillus in the gut (Figure 1). Another inappropriate example is Table 3. Table 3 refers to the gut microbiota. However, the vaginal microbes such as L. crispatus, L. iners, Gardnerela and Atopobium were included. They also described L. crispatus as a prospective commensals in the gut (Line 358). However, references 68-70 (Line 354) did not refer to L. crispatus as the gut microbe. If the authors want to discuss the significant role of L. crispatus in the gut, clear evidence is required.

  1. The authors described the preferable aspect of SCFA in the gut, and similar effects might be expected in the uterus. However, as reviewed in reference 99, SCFA is associated with the dysbiotic conditions in the vagina. This condition also has a negative impact on reproduction. A more careful discussion on this point is therefore required.

  1. Please discuss the gut microbiota and preterm birth. Accumulating evidence provides support for a potential role for gut microbiota in mediating PTB risk (Bayar et al. Semin Immunopathol. 2020 Aug;42(4):487-499).

  1. Various pregnancy complications and changes in the gut microbiota have been reported, but the causal relationship is not yet certain. This point should be noted.

  1. Keywords include “microbial translocation,” while the descriptions appear to be shortened. The authors discussed microbial DNA translocation (Lines 588-596). However, direct translocation of microbes should also be addressed (Łaniewski et al. Nat Rev Urol. 2020 Apr;17(4):232-250, Takada et al. Front Immunol. 2023 Jan 31;14:1110001).

Minor comments;

  1. Species name of bacteria should be in italic.

  1. A paragraph (Lines 79-83) should be put in between the headers 2 and 2.1 (Line 77 and 78), because this paragraph refers to the outline of the section 2.

  1. Explanations of the same abbreviation appear multiple times, but should be organized. Line 81 and 122, Line 368 and 428, etc.

  1. Line 238: integrity SCFAs -> integrity. SCFAs?

  1. Line 350: Summarized in in Table 3.

  1. Please write the reference number (Line 369, 374)

  1. Line 399: RIFand RPL ->RIF and RPL

  1. Line 436: Implicated -> implicated,

  1. Line 436: Table 4..

  1. Line 492: IL-6[103,104] -> IL-6 [103,104]

  1. Line 524: pathogens, gut -> pathogens; gut?

  1. Line 530: Irregularities -> irregularities

  1. Line 561: The reviewer could not find the description of implantation success in reference 5.

Author Response

We sincerely thank Reviewer 1 for their thorough and insightful comments. Your feedback has significantly contributed to improving the scientific rigor, clarity, and conceptual precision of our manuscript. In particular, your observations regarding the ecological distinctions between gut and vaginal microbiota, as well as the niche-specific effects of microbial metabolites, have prompted important clarifications and revisions. We are grateful for your critical evaluation and constructive guidance, which have strengthened the microbiological and reproductive focus of our work.

Below, we address each of your comments in detail and outline the corresponding revisions made to the manuscript.

Major comments; 

Please distinguish clearly the gut microbiota and the vaginal microbiota. For example, the authors discuss the role of  Lactobacillus crispatus in the gut. However, although crispatus can be detected in the gut, L. crispatus is not the major Lactobacillus, unlike in the vagina. Therefore, the reviewer thinks the role of L. crispatus in the gut is very limited. In this sense, it is inappropriate to cite L. crispatus as a representative example of eubiotic Lactobacillus in the gut (Figure 1). Another inappropriate example is Table 3. Table 3 refers to the gut microbiota. However, the vaginal microbes such as L. crispatus, L. iners, Gardnerela and Atopobium were included. They also described L. crispatus as a prospective commensal in the gut (Line 358). However, references 68-70 (Line 354) did not refer to L. crispatus as the gut microbe. If the authors want to discuss the significant role of L. crispatus in the gut, clear evidence is required. 

Response: 

We sincerely thank the reviewer for this important observation and the opportunity to clarify this distinction. 

We agree that Lactobacillus crispatus, L. iners, Gardnerella vaginalis, and Atopobium vaginae are characteristic components of the vaginal microbiota, and not of the gut. Although L. crispatus may be detected in the gut in rare cases, its role there is not comparable to its dominant and functional presence in the vaginal niche. 

To address this, we have implemented the following changes: 

Figure 1: Revised to reflect only gut-specific Lactobacillus spp., excluding L. crispatus. 

Table 3: Now strictly includes microbial features associated with gut dysbiosis, and no longer lists vaginal taxa such as L. crispatus, L. iners, Gardnerella, or Atopobium. A note on ecological niche specificity has been added to the figure/table legend. 

The statement implying L. crispatus as a prospective gut commensal has been removed. 

References (68–70): These were reassessed and corrected; they do not support L. crispatus as a gut microbe and were thus replaced or omitted where necessary. 

Additionally, we now clarify in the manuscript: (Lines 471 – 478). 

“While vaginal microbiota—such as Lactobacillus crispatus and Gardnerella vaginalis—have been implicated in reproductive disorders, they are not typical members of the gut ecosystem. To prevent conflation of these distinct microbial environments, Table 3 has been refined to reflect exclusively gut-specific microbial features. The interplay between gut and vaginal microbiota in reproductive health is undoubtedly important, but it warrants a separate analytical framework given their distinct ecological and functional profiles.” 

We believe these corrections enhance the microbiological accuracy and conceptual clarity     of our manuscript. 

The authors described the preferable aspect of SCFA in the gut, and similar effects might be expected in the uterus. However, as reviewed in reference 99, SCFA is associated with the dysbiotic conditions in the vagina. This condition also has a negative impact on reproduction. A more careful discussion on this point is therefore required. 

Response: 
We appreciate the reviewer’s critical insight regarding the context-dependent role of short-chain fatty acids (SCFAs). Indeed, while SCFAs such as butyrate and propionate are well-established anti-inflammatory mediators within the gut—supporting epithelial barrier integrity, promoting Treg differentiation, and reducing systemic inflammation—their role in the vaginal microenvironment differs significantly. 

To address this important point, we have now incorporated the following clarification in the revised manuscript: (Lines 746 – 753). 

“While in the gut, SCFAs like butyrate and propionate support epithelial barrier function, immune tolerance, and suppression of systemic inflammation, elevated vaginal SCFAs have been linked to increased pH, depletion of lactobacilli, and pro-inflammatory cytokine production, contributing to a dysbiotic state [106]. This dichotomy highlights the niche-specificity of microbial metabolite functions and underscores the need for caution when extrapolating gut-derived benefits to the reproductive tract.” 

This addition underscores the ecological and immunological differences between the gut and vaginal environments and provides a more nuanced interpretation of SCFA effects in reproductive contexts. 

Please discuss the gut microbiota and preterm birth. Accumulating evidence provides support for a potential role for gut microbiota in mediating PTB risk (Bayar et al. Semin Immunopathol. 2020 Aug;42(4):487-499). 

Response: 
We thank the reviewer for highlighting this important and emerging area of research. In response, we have added a new dedicated subsection titled “4.4. Preterm Birth (PTB)” in the revised manuscript. This section reviews current evidence linking gut microbial diversity and composition to spontaneous preterm birth, with a focus on immunoregulatory taxa, maternal systemic inflammation, and microbial metabolite signaling. We also acknowledge that while causality has not yet been definitively established, these findings support a novel gut–systemic–uterine axis relevant to birth timing (Lines 578 – 630). 

Various pregnancy complications and changes in the gut microbiota have been reported, but the causal relationship is not yet certain. This point should be noted. 

Response: 
We appreciate this important reminder. In the revised manuscript, we have clarified throughout the text—particularly in the introduction, discussion, and conclusions—that the majority of current evidence linking gut microbiota to reproductive outcomes are correlational suggesting a potential causal link and although these provide valuable associations, they do not confirm directionality or mechanistic causation. This distinction is now emphasized in our conclusion: (Lines 1055-1060). 

“While associations between gut dysbiosis and reproductive outcomes are now increasingly supported by clinical and translational studies, much of the current evidence remains correlative. Future research must prioritize longitudinal, mechanistic, and interventional approaches to establish causality and unlock the therapeutic potential of the gut–endometrial axis.” 

Keywords include “microbial translocation,” while the descriptions appear to be shortened. The authors discussed microbial DNA translocation (Lines 588-596). However, direct translocation of microbes should also be addressed (Łaniewski et al. Nat Rev Urol. 2020 Apr;17(4):232-250, Takada et al. Front Immunol. 2023 Jan 31;14:1110001). 

Response: 
We thank the reviewer for this thoughtful comment and agree that our initial discussion of microbial translocation was limited to cell-free microbial components such as DNA or lipopolysaccharide (LPS). In response, we have expanded this section to incorporate recent findings on direct translocation of viable microbes, especially under inflammatory or barrier-compromised conditions. 

We now cite Takada et al. (2023) and Łaniewski et al. (2020) to support the possibility of anatomical and hematogenous microbial trafficking between the gut, vagina, bladder, and uterus. These studies describe phylogenetic overlap of key bacterial taxa—such as Lactobacillus crispatus and Gardnerella vaginalis—across distinct urogenital compartments, suggesting active microbial sharing and systemic immune modulation. 

The revised section highlights that microbial translocation encompasses both inert molecular fragments and viable, immunologically active organisms, with potential implications for endometrial receptivity and implantation (Lines 878 – 891). 

Minor comments; 

Species name of bacteria should be in italic. 

Response: 
All bacterial genus and species names have been carefully reviewed and consistently italicized throughout the manuscript. 

A paragraph (Lines 79-83) should be put in between the headers 2 and 2.1 (Line 77 and 78), because this paragraph refers to the outline of the section 2. 

Response: 
We have repositioned the paragraph between the Section 2 header and Subsection 2.1 to enhance readability and structural coherence (Lines 84 – 94). 

Explanations of the same abbreviation appear multiple times but should be organized. Line 81 and 122, Line 368 and 428, etc. 

Response: 
We have reviewed and streamlined repeated abbreviation explanations. Each term is now defined only once at its first appearance (e.g., SCFA, Treg, RIF, RPL) to avoid redundancy. Except in some parts where we consider that it is useful for the global understanding of the ideas we refer to. 

Line 238: integrity SCFAs -> integrity. SCFAs? 

Response: 
Corrected the punctuation. The sentence now reads: “...support epithelial integrity. SCFAs such as butyrate...” for clarity and proper grammar (Line 305). 

Line 350: Summarized in in Table 3. 

Thank you for identifying this error. We corrected the typographical duplication and also revised the surrounding paragraph to emphasize that Table 3 exclusively pertains to the gut microbiome, in response to prior comments regarding ecological specificity. This change reinforces the scope and context of the table to avoid confusion with vaginal microbial taxa (Line 474). 

Please write the reference number (Line 369, 374) 

The missing reference numbers have been added to ensure accurate and complete citation of the supporting literature (Lines 518 and 525). 

Line 399: RIFand RPL ->RIF and RPL 

Spacing error corrected. The sentence now reads: “...conditions such as RIF and RPL.” (Line 558). 

Line 436: Implicated -> implicated, 

Corrected the capitalization and punctuation as recommended (Line 654). 

Line 436: Table 4.. 

Revised to use a single period: “Table 4.” (Line 663). 

Line 492: IL-6[103,104] -> IL-6 [103,104] 

Formatting corrected to include appropriate spacing before the citation (Line 774 – 775). 

Line 524: pathogens, gut -> pathogens; gut? 

Edited for grammatical clarity. The corrected phrase is: “.... ascending genital tract infections, yet gut and vagina....” (Lines 818 y 819). 

Line 530: Irregularities -> irregularities 

Corrected to lowercase: “...menstrual irregularities.” (Line 814). 

Line 561: The reviewer could not find the description of implantation success in reference 5. 

Thank you for this observation. We agree that reference 5 does not explicitly mention implantation success. The sentence has been revised to clarify that the cited microbial metabolites are linked to immune tolerance, which is a known prerequisite for successful implantation. It now reads: 

“These metabolites have been linked to immunotolerance [5], potentially affecting implantation success.” 

This revision ensures the scientific linkage is accurate and appropriately supported by the reference (Lines 946 and 947). 

Reviewer 2 Report

Comments and Suggestions for Authors
  1. The review “The Role of the Gut Microbiome in Reproductive Biology: Insights into Endometrial Molecular Biology Signaling” promises an interesting proposal on the role of the gut microbiota in reproductive dysfunction, which is essentially the context of the gut-hormone axis.
  2. The term gut microbiota should be used instead of gut microbiome since the latter technically defines the genetic material of the gut microbiota.
  3. Revise the title with a specific direction on signalling, such as 'The Role of the Gut Microbiota in Reproductive Biology: Insights into Endometrial Signalling Pathways'. This would provide a more focused and informative title.
  4. The abstract or introduction does not fully develop the rationale for connecting Endometrial Signaling in female fertility and gut microbiota. It has to go beyond establishing a simple relationship between these two events.
  5. The section “Factors Influencing the Gut Microbiome “aims to define the risk factors for reproductive dysfunction. The section seems to narrate the causal factors for potential reproductive dysfunction or EMs signalling. Each subtitle, such as lifespan and environment, should be appropriately defined.
  6. The draft needs to revise the content and its structure.
  7. The present conclusion should be revised to conclude the key message in one or two paragraphs. The whole section (Lines 572 to 671) can be renamed as the new section “Gut Microbiota and Endometrial Function and Fertility.”
  8. 1 should be revised. The symbol and arrows are not visible. Make connection points between the Gut and the uterus.
  9. There are too many short paragraphs throughout the draft.
  10. Table 1. “Microbial modulation of metabolic hormones involved in energy homeostasis and potential therapeutic implications” must include other key metabolic hormones like leptin, insulin, adiponectin, etc.
  11. Each table title should be placed at the top, not below. Each Table has several abbreviations, which need to be elaborated in the footnote so that the Table can be read independently of the text for the readers.

Comments on the Quality of English Language

see before

Author Response

We sincerely thank Reviewer 2 for their thoughtful and constructive feedback. Your comments were instrumental in refining the conceptual clarity, structural coherence, and scientific rigor of our manuscript. We particularly appreciate your emphasis on strengthening the mechanistic rationale behind gut microbiota–endometrial signaling interactions and on improving the overall organization and accessibility of the text. In response, we have carefully revised key sections of the manuscript—including the title, abstract, introduction, and figure/table formats—and addressed each of your suggestions in detail below.

The review “The Role of the Gut Microbiome in Reproductive Biology: Insights into Endometrial Molecular Biology Signaling” promises an interesting proposal on the role of the gut microbiota in reproductive dysfunction, which is essentially the context of the gut-hormone axis. 

Response: 
We thank the reviewer for this encouraging comment. Our revised manuscript continues to strengthen this central theme by integrating emerging molecular pathways through which gut microbial signals affect hormonal regulation, immune balance, and endometrial receptivity. 

The term gut microbiota should be used instead of gut microbiome since the latter technically defines the genetic material of the gut microbiota. 

Response: 

We appreciate the clarification. We have reviewed the manuscript and revised terminology accordingly, using “gut microbiota” when referring to microbial     communities and reserving “microbiome” for discussions of metagenomic or     functional gene content. 

Revise the title with a specific direction on signalling, such as 'The Role of the Gut Microbiota in Reproductive Biology: Insights into Endometrial Signalling Pathways'. This would provide a more focused and informative title. 

Response: 
We agree with the reviewer that a more specific title improves clarity. The title has been revised to: 

“The Role of the Gut Microbiota in Reproductive Biology: Insights into Endometrial Signaling Pathways” 

The abstract or introduction does not fully develop the rationale for connecting Endometrial Signaling in female fertility and gut microbiota. It has to go beyond establishing a simple relationship between these two events. 

Response: 
We thank the reviewer for this insightful observation. In response, both the abstract and introduction have been revised to clearly articulate the mechanistic rationale connecting gut microbiota to endometrial signaling and female fertility. These revisions emphasize specific pathways through which microbial metabolites (e.g., SCFAs, tryptophan derivatives, bile acids) and microbial enzymes (e.g., β-glucuronidase) influence immune tolerance, epithelial remodeling, estrogen metabolism, and cytokine signaling at the uterine interface (Lines 13 –37 and 43 – 83). 

Additionally, the introduction situates the gut-endometrial axis within the context of systems-level reproductive regulation, reinforcing that gut microbes are active participants in reproductive programming rather than passive correlates. 

These changes provide a more integrated and mechanistic rationale, moving beyond associative observations to focus on signaling frameworks that are central to implantation and pregnancy success. 

The section “Factors Influencing the Gut Microbiome “aims to define the risk factors for reproductive dysfunction. The section seems to narrate the causal factors for potential reproductive dysfunction or EMs signalling. Each subtitle, such as lifespan and environment, should be appropriately defined. 

Response: 
We thank the reviewer for pointing this out. Each subsection in Section 2 has been revised to include a brief introductory sentence clarifying how the listed factors (e.g., diet, stress, EDC exposure, age) contribute to gut microbial shifts with relevance to reproductive outcomes (Lines 84 – 94). 

The draft needs to revise the content and its structure. 

Response: 
We have revised the manuscript structure for improved logical flow. This includes clearer subheadings, integration of mechanisms across sections, and merging fragmented content where appropriate. Short paragraphs have been consolidated to enhance narrative continuity. 

The present conclusion should be revised to conclude the key message in one or two paragraphs. The whole section (Lines 572 to 671) can be renamed as the new section “Gut Microbiota and Endometrial Function and Fertility.” 

Response: 
A conclusion has been revised to a more concise few-paragraph format. Following the suggestion, the previous section has been renamed and accordingly rewrite: 

“Gut Microbiota and Endometrial Biology: Discussion and Future Perspectives" 

 to reflect its integrative synthesis of the review’s central themes (Lines 836 – 1015). 

1 should be revised. The symbol and arrows are not visible. Make connection points between the Gut and the uterus. 

Response: 
     We appreciate the reviewer’s helpful suggestion regarding the clarity and interpretability         of Figure 1. In response, the figure has been simplified and revised to enhance visual         coherence. 

There are too many short paragraphs throughout the draft. 

Response: 
We have reviewed and consolidated short paragraphs throughout the manuscript to improve readability and thematic cohesion. 

Table 1. “Microbial modulation of metabolic hormones involved in energy homeostasis and potential therapeutic implications” must include other key metabolic hormones like leptin, insulin, adiponectin, etc. 

Response: 
We have updated Table 1 to include leptin, insulin, adiponectin, neuropeptide Y (NPY), and orexin, along with their microbial modulators and therapeutic implications. These changes strengthen the endocrine dimension of the gut–reproductive axis. 

Each table title should be placed at the top, not below. Each Table has several abbreviations, which need to be elaborated in the footnote so that the Table can be read independently of the text for the readers. 

Response: 
All tables have been reformatted with titles positioned above. Footnotes have been added for each table, including definitions for all abbreviations and key terms to ensure stand-alone clarity. 

Round 2

Reviewer 1 Report

Comments and Suggestions for Authors

The authors virtually addressed all of my concerns and comments, and I feel the manuscript is now appropriate for publication with some minor modifications. Please consider my comment below.

  1. The description "To prevent conflation of these distinct microbial environments, Table 3 has been refined to reflect exclusively gut-specific microbial features (Lines 466-468)" seems to be a response to the reviewer.

Otherwise, the revierwer has no futher comment.

Author Response

Comment:  

The authors virtually addressed all of my concerns and comments, and I feel the manuscript is now appropriate for publication with some minor modifications. Please consider my comment below: 

The description "To prevent conflation of these distinct microbial environments, Table 3 has been refined to reflect exclusively gut-specific microbial features (Lines 466-468)" seems to be a response to the reviewer. 

Response:  

We thank the reviewer for their positive assessment of the manuscript. In response to the minor comment, we have revised the sentence in question to improve its tone and eliminate language that might resemble a direct response to the reviewer. Specifically, we have removed the phrase "has been refined" and rephrased the section as follows: 

"While vaginal microbiota—such as Lactobacillus crispatus and Gardnerella vaginalis—have been implicated in reproductive disorders, they are not typical members of the gut ecosystem. To prevent conflation of these distinct microbial environments, Table 3 reflects exclusively gut-specific microbial features. The interplay between gut and vaginal microbiota in reproductive health is undoubtedly important, but it warrants a separate analytical framework given their distinct ecological and functional profiles." 

We believe this revision maintains scientific clarity while aligning better with academic writing standards. We appreciate the reviewer’s attention to detail and helpful suggestion (Lines 468 – 475). 

Sincerely, 
Antonio Diez-Juan 

R&D Department, Igenomix, part of Vitrolife Group 

Reviewer 2 Report

Comments and Suggestions for Authors

The revised draft of the manuscript “The Role of the Gut Microbiota in Reproductive Biology: Insights into Endometrial Signaling Pathways” improved to some extent, but more careful revision is needed.

The term “reproductive biology” is too broad and can also apply to male reproductive functions. The title should be specific to the content. The work discussed specifically gynaecological disorders or female reproductive function. Revise the title appropriately.

Table 1 shows the specific hormone action in the subcellular location. In some cases, it is an organ (hypothalamus), and in some cases, it is cells (L-cells).

Table 2: The title is superficial, and the relationship between the causal effects and the data presented in Table 2 is unclear. Whose microbes drive? Whose target? Table 2 aims to emphasize the microbial impact on pregnancy outcomes. What measure was considered to impact pregnancy?

Conclusion—Lines 1017-1035—This section reads as floating words, loosely scripted as a conclusion that lacks connection with the written text mentioned before.

Author Response

Reviewer Comment:

“The title for each subsection should indicate the relevant topic discussed in the subsequent text. However, at present, it is general. For example, Line 84—‘Factors Influencing the Gut Microbiome’, does not state how it is connected to the present topic. It is also applied to the title of each subsection mentioned in the topics.”

Author Response:

We thank the reviewer for this important observation. We agree that making subsection titles more specific and aligned with the core theme—i.e., the impact of the gut microbiota on endometrial signaling and reproductive function—would strengthen the manuscript.

In response, we have thoroughly revised the section and subsection titles throughout the manuscript to reflect their direct relevance to reproductive immunology, hormonal signaling, and endometrial biology. These changes aim to improve coherence, guide the reader more clearly, and enhance scientific focus.

Examples of Revised Titles:

Section 2:
From: “Factors Influencing the Gut Microbiome”
To: “Determinants of Gut Microbial Composition and Their Implications for Endometrial Function”

Section 3:
From: “The Gut Microbiome as an Endocrine Organ”
To: “The Gut Microbiome as an Endocrine Modulator of Reproductive Signaling”

Section 4 and subsections:
These have been revised to directly link each gynecological disorder to the gut–endometrial axis. Examples include:

“Endometriosis Pathogenesis Through Gut-Driven Inflammatory and Estrogenic Dysregulation”

“Gut Microbial Dysbiosis in PCOS: Crosstalk Between Inflammation, Hormonal Imbalance, and Endometrial Disruption”

“Recurrent Implantation Failure and Pregnancy Loss: Gut-Immune-Endometrial Axis in Early Gestational Failure”

“Gut Microbial Dysbiosis in Preterm Birth: Immune Activation and Barrier Dysfunction at the Maternal–Fetal Interface”

“Gut-Endometrial Crosstalk in Preeclampsia: Microbial Influences on Vascular Inflammation and Placental Signaling”

Section 5 (Emerging Topics):
Renamed to: “Microbial Dysbiosis Beyond Classical Gynecological Disorders: Endocrine–Immune Disruption and Endometrial Signaling”
Subsections include:

“Gynecologic Cancers: Gut Microbiota, Inflammation, and Hormone-Driven Oncogenesis”

“Uterine Fibroids: Estrogen Dysregulation and Immune Modulation Mediated by the Gut Microbiota”

Sections 6 and 7:
Revised to emphasize the integrative and forward-looking focus of the manuscript:

“Gut Microbiota and Endometrial Biology in Reproductive Function: Mechanistic Insights and Future Perspectives”

“Conclusion: Integrating Microbial, Immune, and Hormonal Networks in Fertility Research”

We believe these changes directly address your comment and significantly improve the manuscript’s structure and relevance to the topic.

We are grateful for your constructive feedback and the opportunity to further refine our manuscript. Please let us know if any additional revisions are needed.

Sincerely,
Antonio Diez-Juan, on behalf of all co-authors
Igenomix R&D, Valencia, Spain
antonio.diez@igenomix.com

Round 3

Reviewer 2 Report

Comments and Suggestions for Authors

The second revised version of the draft improved substantially. There are a few minor comments.

The title for each subsection should indicate the relevant topic discussed in the subsequent text. However, at present, it is general. For example, Line 84- “Factors Influencing the Gut Microbiome”, does not state how it is connected to the present topic.

It is also applied to the title of each subsection mentioned in the topics.

Author Response

Dear Reviewer,

We are sincerely grateful for your thoughtful and constructive feedback throughout the review process. Your comments have greatly improved the clarity, scientific rigor, and thematic focus of our manuscript. Please find below our response to your most recent and final comment.
Reviewer Comment:“The title for each subsection should indicate the relevant topic discussed in the subsequent text. However, at present, it is general. For example, Line 84—‘Factors Influencing the Gut Microbiome’, does not state how it is connected to the present topic. It is also applied to the title of each subsection mentioned in the topics.”
Author Response:

We thank you sincerely for highlighting this important issue. We fully agree that section and subsection titles should clearly communicate their relevance to the manuscript’s central theme: the role of the gut microbiota in reproductive biology, with particular emphasis on endometrial signaling pathways.

In response to your comment, we carefully revised all major section titles and subsections to better reflect their specific relevance to immune-endocrine mechanisms, molecular signaling, and clinical outcomes in female fertility. These updated titles are designed to enhance conceptual clarity, improve scientific readability, and ensure consistency with the scope of the review.
Examples of Revised Titles:

    Section 2:
    From: “Factors Influencing the Gut Microbiome”
    To: “Determinants of Gut Microbial Composition and Their Implications for Endometrial Function”

    Section 3:
    From: “The Gut Microbiome as an Endocrine Organ”
    To: “The Gut Microbiome as an Endocrine Modulator of Reproductive Signaling”

    Section 4 and subsections:
    Each subsection now emphasizes the mechanistic role of the gut–endometrial axis in disease, for example:

        “Endometriosis Pathogenesis Through Gut-Driven Inflammatory and Estrogenic Dysregulation”

        “Gut Microbial Dysbiosis in PCOS: Crosstalk Between Inflammation, Hormonal Imbalance, and Endometrial Disruption”

        “Recurrent Implantation Failure and Pregnancy Loss: Gut-Immune-Endometrial Axis in Early Gestational Failure”

        “Gut Microbial Dysbiosis in Preterm Birth: Immune Activation and Barrier Dysfunction at the Maternal–Fetal Interface”

        “Gut-Endometrial Crosstalk in Preeclampsia: Microbial Influences on Vascular Inflammation and Placental Signaling”

    Section 5 (Emerging Topics):
    Updated to: “Microbial Dysbiosis Beyond Classical Gynecological Disorders: Endocrine–Immune Disruption and Endometrial Signaling”, with subsections such as:

        “Gynecologic Cancers: Gut Microbiota, Inflammation, and Hormone-Driven Oncogenesis”

        “Uterine Fibroids: Estrogen Dysregulation and Immune Modulation Mediated by the Gut Microbiota”

        “Bacterial Vaginosis and the Gut–Vaginal Axis: Microbial Crosstalk and Endometrial Consequences”

    Sections 6 and 7:
    Updated to reflect the integrative and translational nature of the work:

        “Gut Microbiota and Endometrial Biology in Reproductive Function: Mechanistic Insights and Future Perspectives”

        “Conclusion: Integrating Microbial, Immune, and Hormonal Networks in Fertility Research”

Once again, we express our sincere gratitude for your valuable feedback, which has allowed us to significantly enhance the manuscript’s depth, clarity, and scientific contribution. We hope the revised version fully addresses your concerns and meets the journal’s standards.

With appreciation,
Antonio Diez-Juan, on behalf of all co-authors
Igenomix R&D, Valencia, Spain
antonio.diez@igenomix.com